# The Oxidative Stress Markers of Horses—the Comparison with Other Animals and the Influence of Exercise and Disease

**DOI:** 10.3390/ani10040617

**Published:** 2020-04-03

**Authors:** Saori Shono, Azusa Gin, Fumiko Minowa, Kimihiro Okubo, Mariko Mochizuki

**Affiliations:** 1Department of Applied Science, School of Veterinary Nursing and Technology, Nippon Veterinary and Life Science University, Tokyo 180-8602, Japan; sshono@nvlu.ac.jp (S.S.); ginnnn525@yahoo.co.jp (A.G.); 2Minowa Horse Clinic, Tokyo 144-0033, Japan; hgf-mino@y9.dion.ne.jp; 3Department of Otolaryngology, Graduate School of Medicine, Nippon Medical School, Tokyo 113-8602, Japan; ent-kimi@nms.ac.jp

**Keywords:** diacron-reactive oxygen metabolite, biological antioxidant potential, horse, oxidative stress index

## Abstract

**Simple Summary:**

Although oxidative stress is detrimental in biological systems, direct analysis of the active oxygen species that causes the stress has been difficult at a clinical level. In the current study, we analyzed the levels of diacron-reactive oxygen metabolites (d-ROMs) and biological antioxidant potential (BAP) in the serum of horses. These are easy to measure and they provide information on the level of oxidative stress in an animal. The mean d-ROM level in horses was higher than those in dogs or dairy cattle, and the levels in horses can be used to distinguish those with a disease.

**Abstract:**

Diacron-reactive oxygen metabolite (d-ROM) and biological antioxidant potential (BAP) levels in the serum of horses were measured (ponies, *n* = 15; thoroughbred, *n* = 31; other full-sized horses, *n* = 7). The mean d-ROM levels in horses were significantly higher (*p* < 0.001) than those in dairy cattle (*n* = 25) and dogs (*n* = 31). However, d-ROM levels in horses were lower than the standard levels reported in humans. When d-ROM and BAP levels were plotted graphically, the points for horses with a disease (ringbone in 1 Japanese sports horse, cellulitis in 1 thoroughbred, melanoma in 1 Lipizzaner) fell outside the group of points for other (non-diseased) horses. A similar separation was seen (using data from other authors) for a horse with *Rhodococcus equi*, a horse following castration surgery, and a mare following delivery. These results, comparing horses, other animals, and humans, are interesting from the standpoint of comparative medicine, and they contribute to the sparse literature available on d-ROM and BAP levels in animals. Because the level of d-ROM and BAP levels were changed depending on the situation of health, those indexes are promising as indices of health in horses.

## 1. Introduction

Oxidative stress is detrimental to biological systems. It occurs when the balance between reactive oxygen species and antioxidants is disturbed. In humans, it is well known that oxidative stress is associated with various diseases, including diabetes [1], obesity [2], inflammation, and cancer [3], and with lifestyle habits such as fitness activities [4], alcohol consumption [5] and smoking [6]. There have also been many studies in various animals using indexes related to oxidative stress and antioxidant activity such as 8-OHdG [7] and lipid peroxidation [8]. The aims of those studies were to cover a lot of ground, for example, animal welfare [9,10], produce of value-added animal product [11], efficient produce of animal product [7], and estimation of disease [12].

Because of their high activity and short half-life, direct analysis of the reactive oxygen species that cause oxidative stress has been difficult at the clinical level; but more recently, the equipment has improved, and the measurement of oxidative stress levels is relatively easy. The recent method involves measuring hydroperoxides (ROOH) levels as metabolic products to determine the level of oxidative stress as diacron-reactive oxygen metabolites (d-ROMs). Because it is easy to maintain and operate, the measuring equipment has been used in relation to human diseases in clinical practice, with reports appearing on cardiovascular disease [13], high-density lipoprotein [14], and pediatric oncology [15], although there are few published studies using this instrument on animals.

In the study related animals, the d-ROMs concentration has been used as an index of stress. Fazio et al. [9] reported that higher d-ROMs concentration was observed in dogs due to road transportation. In the study using dogs protected in a shelter, d-ROMs concentration of serological negative dogs was significantly higher than that of serological positive dogs [10]. Oxidative stress is also monitored for health management in athletes [16]. As with athletes, racehorses and competition horses experience high levels of physical exercise, and although knowledge of the status of d-ROMs levels in the serum of horses might provide important information on their condition, there is little research published on the same. Further, not only for horses, there is also little information about the data of other animals such as dogs. To know the character of data of d-ROMs in the serum of horses, the comparison with the results of other animals is thought to be interesting. Thus, in the present study, the concentration of d-ROMs in the serum of horses is compared with that in other animals (dairy cattle and dogs) for the influence of exercise and disease on d-ROM levels. Then, the influence on d-ROMs of exercise load and diseases in horses is investigated. Finally, we investigate antioxidant capacity by measuring biological antioxidant potential (BAP).

## 2. Materials and Methods

### 2.1. Animals and Sample Preparation

The study was approved by the ethical policies of experimental animals of Nippon Veterinary and Life Science University (approval number, daily cattle: 28K-20, S28K-20; horse: 27S-8, 28K-19 and 29; dogs: 28S-9, S28-9). Serum samples were obtained from horses, dairy cattle, and dogs. The serum samples of horses were collected at an equestrian club in Shizuoka Prefecture in Japan on 19 August 2015 during separate research (ponies; *n* = 12, thoroughbreds; *n* = 26) (Red symbols in Figure 1, Figure 2 and Figure 3). Three ponies and four thoroughbreds in this club with decreased physical activity (mean value of these 7 animals was indicated in [17]) are shown by black symbols in Figure 2. The information on the horses (such as sex and age) were reported previously [17]. A total of 8 full-sized horses fed in the same club are newly added in this study, as follows: thoroughbred (1 gelding, age 20 y), Selle Français (2 females, average age 12 y; 2 geldings, average age 13.5 y), KWPN (1 gelding, age 15 y), Anglo-Arabian (1 gelding, age 22 y), DWB (1 gelding, 23 y). Those 8 full-sized horses are manifested in red symbols in Figure 1, Figure 2 and Figure 3 (thoroughbred, circle; others, triangle). Those samples were also collected on 19 August 2015. Horses at an equestrian club in Shizuoka Prefecture that were suffering from disease were also sampled: Japan sports horse species (gelding, 10 y, ringbone), thoroughbred (gelding, 26 y, cellulitis), and a Lipizzaner (gelding, 30 y, melanoma) (see black symbols in Figure 3). On the day of sampling, the horses were fed at 6:00 a.m. and then pastured between 7:30 and 10:00 a.m. Samples were collected between 10:30 a.m. and 12:00 noon before the horses were used for horseback riding. Serum was collected from a total of 27 dairy cattle consisting of Holstein Friesian (*n* = 25, average age 52.72 ± 3.20 m; green circle symbols in Figure 1), Jersey (*n* = 2, average age 59.97 m; green square symbol in Figure 1) and Brown Swiss (*n* = 1, age 62.2 m; green triangle symbols in Figure 1) at a farm in Yamanashi Prefecture attached to the Nippon Veterinary and Life Science University. The samples from the dairy cattle were collected during another investigation related to animal-assisted education on a farm [18]. In the current day of sampling on 9 May 2016, cattle were fed at 5:00 a.m. The milking was terminated at 6:30 a.m. Sampling was conducted between 10:00 and 11:00 a.m. Blood samples (20 mL) from the horses and dairy cattle were collected into vacuum blood collection tubes (Venoject II, Terumo Corporation, Tokyo, Japan). After coagulation had taken place, the samples were centrifuged at 3000 rpm (1468× *g*) for 15 min. The supernatants were collected into sample tubes (BM Equipment Co., Ltd., Tokyo, Japan). Except for the horses having disease used in Figure 3, all animals, including cattle, dogs, and horses, were not in treatment due to disease.

Frozen serum or plasma of dogs after examination at an animal hospital in Tokyo were transported immediately to the laboratory of Nippon Veterinary and Life Science University. Those samples of dogs were residue samples of the health exam in an animal hospital. Although the time of blood sampling varied, the animal hospital has recommended to at least eat nothing 8 h before the health exam. The sample of dogs was collected between April and July 2017. The time and day of sampling were not consolidated due to a health exam. A total of 31 samples were obtained from toy poodles. Information about these dogs is summarized in Table 1. The serum samples in tubes were stored at −30 °C in a freezer (Panasonic Healthcare Co. Ltd., Tokyo, Japan) until analysis.

### 2.2. Data on Horses from Previous Studies

Some of the data on the horses in this study were taken from previous studies: the data obtained from endurance horses before (green square symbols in Figure 2) and after (green circle in symbols Figure 2) an endurance race [19], and data for thoroughbreds before and after exercise using a treadmill [20] were used as a model for healthy horses that experienced increased exercise.

Data from some special cases were also included: horses with *Rhodococcus equi* [21], after delivery [22], and after castration surgery [23] (black symbols in Figure 3). 

### 2.3. Measurement

Because the retention period of stored samples is recommended until 1 year under less than −20 °C in the manual provided by the manufacturer, the stored samples were analyzed within a time limit of the storage period. On the day of analysis, samples were transferred to an ice chamber to thaw.

The concentration of d-ROMs and BAP in serum was analyzed using a free radical analyzer (Free Carpe Diem, Diacron International srl., Grosseto, Italy). The oxidative stress index (OSI) was calculated using the formula (d-ROM / BAP × 100).

### 2.4. Statistical Methods

Data are presented as mean ± standard error of the mean. Analyses were performed using the Japanese language version of Lotus 2001 software (Microsoft Japan Co., Ltd., Tokyo, Japan) and Excel 2016 (Microsoft Japan Co., Ltd., Tokyo, Japan). The normality of data has been checked by the Kolmogorov–Smirnov test and the Shapiro–Wilk test. Because the data in the present study has shown a mixture of non-normal and normal and distribution, significant differences in elemental concentrations were tested for using the Kruskal–Wallis test. In addition, the Dunn test was used as a post hoc test. The differences in 2 variates were tested for using the Mann–Whitney *U*-test. Those statistical tests were performed using the Japanese language version of IBM statistics 19 (IBM Japan Ltd., Tokyo, Japan) and Bell curve (Social Survey Research Information Co., Ltd., Tokyo, Japan).

## 3. Results

### 3.1. Comparison between Horses and Other Animals

The concentration of d-ROMs and BAP in the serum of horses and other animals are shown in Table 2 and Figure 1. The value for horses was significantly higher than for both dairy cattle (*p* < 0.001) and dogs (*p* < 0.001). However, horse d-ROM values in the present study were lower than the reference value for d-ROMs in humans.

The mean serum BAP concentrations in the horses, dairy cattle and dogs were 2237.70, 2622.09 [18], and 2227.81 μmol/L, respectively. The value for dairy cattle was significantly higher than for both horses (*p* < 0.001) and dogs (*p* < 0.001), but there was no significant difference (*p* = 0.96) between values for dogs and horses. Due to the higher d-ROM value, the OSI for horses was significantly higher (*p* < 0.001) than for both dogs and dairy cattle.

### 3.2. Comparisons Among Horses

The concentrations of d-ROMs and BAP in the serum of riding horses in an equestrian club and race and endurance horses are shown in Figure 2. There were seven animals with decreased physical activity due to old age or exercise limitation (black symbols in Figure 2) for which the BAP level (mean: 1758.30 μmol/L [17]) was lower than that of normal horses. In contrast, d-ROM and BAP levels tended to be higher in racehorses (green symbols in Figure 2) and endurance horses (blue symbols in Figure 2). When horses were considered in 3 categories, horses in equestrian club (*n* = 46), before endurance race (*n* = 10) and simulated race (*n* = 2), after endurance race (*n* = 10) and simulated race (*n* = 2), a significant difference (*p* < 0.01) were obtained in d-ROMs concentrations in the serum of horses in an equestrian club than of other two groups. A similar significant difference (*p* < 0.001) was also obtained in the investigation of BAP. The significant difference (*p* < 0.001) between before race and after race was obtained on only the BAP level.

Values for d-ROMs and BAP in the serum of horses suffering from a disease, such as ringbone, melanoma, *Rhodococcus equi* [21], or cellulitis, were distributed away from the values for normal horses, as were the values for a horse after delivery [22] and a horse after castration surgery [23] (Figure 3). There is a significant difference (*p* < 0 001) between the mean concentration of d-ROMs in the serum of horses having disease (*n* = 4) and that of horses having no disease (*n* = 46). A similar tendency (*p* < 0.05) was also obtained on the investigation of BAP.

## 4. Discussion

In the first experiment in the present study, we compared d-ROM and BAP concentrations in serum of horses versus dogs and dairy cattle. In the study using healthy Labrador dogs, it was reported that the concentration of d-ROMs and BAP were 73.9 ± 8.78 UCARR and 2446 ± 585 μmol/L, respectively [24]. The results of d-ROMs in the study [24] were lower than those in our study. We also analyzed d-ROMs and BAP concentration in serum of Labrador dogs (*n* = 4), the data (d-ROMs; 70.25 UCARR, BAP; 2121.35 μmol/L) were similar with the study by Pasquini et al. [24]. It was thought that there are differences depending on the breed. In the previous study related to the dairy cattle (*n* = 74), it was reported that d-ROMs and BAP concentration was 158 ± 23.0 UCARR and 2558 ± 108.5 μmol/L, respectively. The result of d-ROMs was slightly lower than that of the previous study [25].

The d-ROM concentrations in horses were significantly higher (*p* < 0.001) than in the other animals. In reports on human health, d-ROM levels are increased due to lifestyle habits such as fitness activities [4], alcohol consumption [5], and smoking [6]. Clearly, because animals are not in the habit of smoking or drinking, the higher d-ROM levels in horses must be influenced by other factors. Because the amount of exercise performed by dairy cattle and dogs was thought to be obviously less than that of horses, the results suggest that the exercise amount is one reason for the higher d-ROM levels in horses. The density of oxygen-carrying erythrocytes in the blood of horses is known to be greater than in other animals such as dogs and cattle [26]. Further, the concentration of hemoglobin in the blood is increased depending on exercise intensity due to an increase in the blood cell:volume ratio. Thus, it is possible that horses maintain their blood oxygen at the higher level required for exercise [26]. Because the level of d-ROMs after a race in the blood of athletes in various sports such as a race of duathlon [27], mountain biking [16], who consume more oxygen, is invariably higher than before the race, the basis of the higher d-ROM concentration in serum of horses may be for the same reason.

Then, we compared d-ROM concentration in the serum of horses owned by an equestrian club against race and endurance horses. The d-ROM levels of the race and endurance horses were relatively high and focused, while those of the horses in the equestrian club were more widely distributed. The horses from the equestrian club varied in age and breed, and the amount of exercise they received differed depending on age and role, and it is probably for these reasons that their d-ROM levels fell over a wider range. The BAP concentration in serum of race and endurance horses was higher than in the other horses in this study. Because a higher BAP level was also reported in the other study using thoroughbred racehorses [28], it is suggested that the differences in exercise significantly influence the level of OS biomarkers. It has been reported [29] that the plasma BAP concentration in active older people is significantly higher than in the inactive elderly. Further, in a study of triathletes of the duathlon (running, cycling, running), d-ROM levels after the first run were significantly higher than levels observed before the race. However, because BAP levels also increased, the overall effect was that OSI gently decreased during the race [28]. The mean OSI in horses in the present study (range: 5.54–7.19) was similar to that of endurance horses [19], although considerable individual variability (range: 4.84–9.42) was observed in the horses in the present study. It has been suggested [30] that there is a significant negative correlation between age and BAP concentration in the serum of older adults. Further, a significant correlation was also observed between the ratio between d-ROMs and BAP. Thus, it is possible that BAP and OSI levels decrease with age. In the study using thoroughbred racehorses [28], it was also suggested that the higher BAP level and lower OSI are reported in low age of female horses. However, the decreasing trend of BAP level in male horses is not marked than that of females in this report. Because this study suggests that there is a gender difference in the level of BAP, an investigation depending on the gender difference is necessary in further study. Because the age of the horses at the equestrian club was obviously higher than that of the dairy cattle, the higher OSI, due to lower BAP levels in the serum of the horses, was thought to be age-related.

The concentrations of d-ROMs and BAP in the serum obtained from horses whose circumstances were other than normal were investigated in the present study. The horse diseases witnessed (ringbone, cellulitis, and melanoma) were chronic and serious illnesses, especially for the horse with melanoma, which was in a terminal condition. In humans, it has been suggested that changes in d-ROM levels are indicative of disease conditions [1,2,3,4,5,6]. Results from the present study also suggest that this index could be useful for the early detection of disease and/or disease conditions in horses. d-ROM levels for horses having no disease but having experienced castration surgery or delivery were also distributed outside the normal range, indicating that it is not only disease that is reflected in d-ROM and BAP levels in serum. Although the synovial fluid was used in the study, d-ROMs and OSI levels in the synovial fluid of the arthritis group were significantly higher than the control group [12]. Because the study of dairy cattle also found higher concentrations of d-ROMs and lower concentrations of BAP in dairy cattle having left displacement of abomasum, it was suggested, when compared to the control group [25], that several diseases are thought to influence d-ROMs concertation of animals. It has also been suggested that d-ROMs concentration rises under stress such as transportation [9] and shelter [10] in the studies using dogs. Because it is well known from long ago that there are horses feeling stress [31], the influence of stress should be also investigated on the horses. Because there are few reports to date on d-ROM and BAP levels in horses, further study would be necessary to describe more precisely how d-ROM and BAP levels change under unusual conditions. Nevertheless, measurement of d-ROM and BAP levels are a useful indicator of health in horses.

## 5. Conclusions

The serum d-ROM and BAP levels in horses differ from that in dogs and dairy cattle. This study suggests that d-ROM and BAP levels may be distributed over a wide range depending on the species. The data is interesting from the viewpoint of comparative medicine. Knowledge of d-ROMs and BAP in horses may help in studies on horses. Further detailed studies investigating more animal species would be desirable.

## Figures and Tables

**Figure 1 animals-10-00617-f001:**
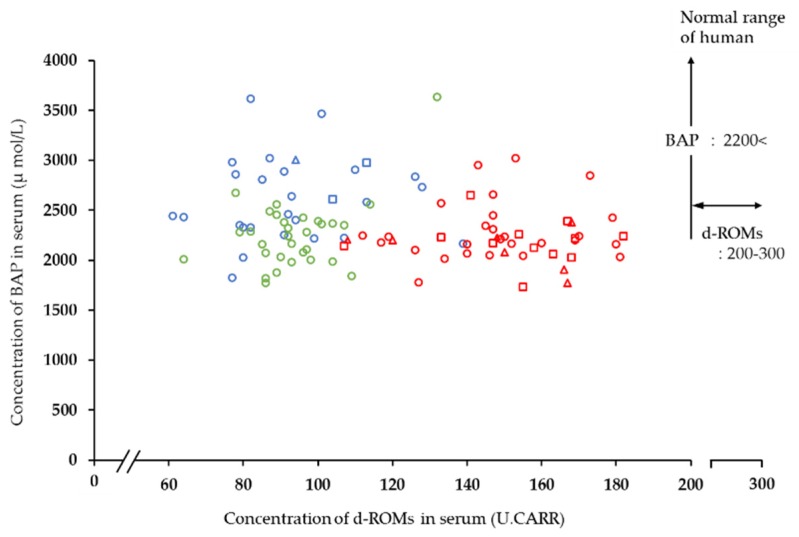
Comparison between diacron-reactive oxygen metabolite (d-ROM) and biological antioxidant potential (BAP) concentrations in serum of horses in an equestrian club in Shizuoka Prefecture and that of other animals (dairy cattle and dogs). Red symbols: horses in an equestrian club in Shizuoka Prefecture used in our study (circle: thoroughbred; square: pony; triangle: full-sized horses other than thoroughbred). Blue symbols: dairy cattle (circle: Holstein Friesian; triangle: Brown Swiss; square: Jersey). Green symbols: dogs.

**Figure 2 animals-10-00617-f002:**
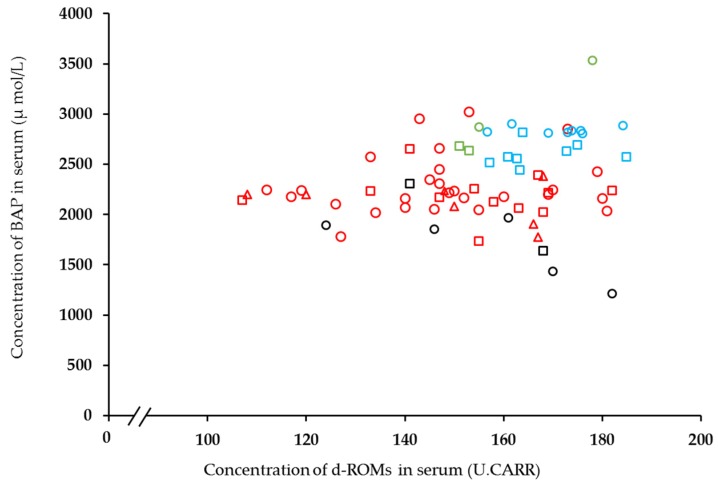
Comparison between d-ROM and BAP concentrations in serum of horses in an equestrian club in Shizuoka Prefecture and that of race and endurance horses. Red and black symbols: horses in an equestrian club in Shizuoka Prefecture in our study (circle: thoroughbreds; square: ponies; triangle: full-sized horses other than thoroughbreds). Black symbols indicate horses with decreased physical activity (mean value of these 7 animals was indicated in [17]). Blue symbols: endurance horses [19] (square: before a race; circle: after a race). Green symbols: racehorses (thoroughbreds) (square: before simulated race; circle: after simulated race) [20].

**Figure 3 animals-10-00617-f003:**
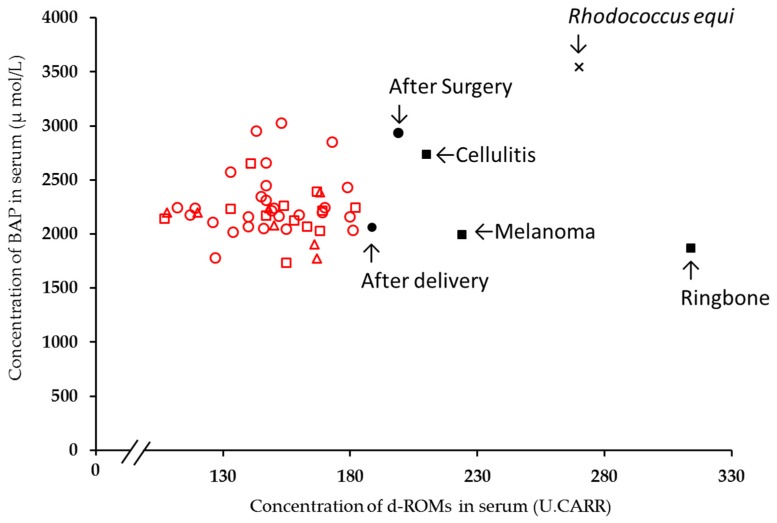
Comparison between d-ROM and BAP concentrations in serum of horses in an equestrian club in Shizuoka Prefecture and that of horses suffering disease or other stress. Red symbols: horses in an equestrian club in Shizuoka Prefecture in our study (circle: thoroughbreds; square: ponies; triangle: full-sized horses other than thoroughbreds). Black symbols: horses suffering disease or other stress (square: horses in our study; other symbols: data from previous reports [21,22,23]).

**Table 1 animals-10-00617-t001:** Information on the dogs used in the present study.

Breed	Sex	*n*	Age (Months)	Weight (Kg)
Toy poodle	Female	6	47.17 ± 26.52	4.43 ± 0.80
	Neuter	14	47.14 ± 11.37	5.38 ± 0.46
	Spay	11	46.18 ± 13.37	3.85 ± 0.42
Total		31	46.81 ± 8.31	4.65 ± 0.29

Data are presented as mean ± standard error of the mean. *n*: number of samples.

**Table 2 animals-10-00617-t002:** d-ROM and BAP levels in animal sera and oxidative stress index (OSI).

Animal	*n*	d-ROMs (U.CARR)	BAP (μmol/L)	OSI	Figure
**Dog**	31	93.74 ±2.20 ^b^	2258.38 ± 62.34 ^b^	4.20 ± 0.11 ^b^	Green symbols in Figure 1
**Horse**	46	149.24 ± 2.92 ^a^	2237.70 ± 39.71 ^b^	6.76 ± 0.18 ^a^	Red symbols in Figure 1
**Dairy cattle**	28	93.48 ± 3.42 ^b^	2622.09 ± 78.09 ^a^	3.65 ± 0.17 ^b^	Blue symbols in Figure 1

d-ROMs: diacron-reactive oxygen metabolites; BAP: biological antioxidant potential; OSI: oxidative stress index calculated as (d-ROMs/BAP × 100). *n*: number of samples. Different lowercase superscript letters in a column indicate significant differences (*p* < 0.001). Data are presented as mean ± standard error of the mean. The mean concentration of d-ROMs and BAP were indicated in our previous report [18].

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
