# Peer review of "The Oxidative Stress Markers of Horses—the Comparison with Other Animals and the Influence of Exercise and Disease"

_animals, 2020, doi:10.3390/ani10040617_

Round 1

Reviewer 1 Report

Dear Authors,

thank you for the reply to suggestions. Please find below some further comments to the revised version of the paper

Introduction:

Line 61: I suggest to change “comparing” with “comparison”

Line 63: I suggest to delete “to establish their characteristic d-ROM levels”. I suggest to rephrase as “the influence of exercise and disease on d-ROM levels…”

Materials and Methods:

As a general comment you should begin with the description of inclusion criteria and sample size and then decribe the method used

Lines 70-71: What does the following sentence mean? “The mean values for d-ROMs and BAP in ponies and thoroughbreads”. I believe something is missing. Moreover these are results. I suggest to delete

 Lines 71-72: again, these are results

Lines 81-82: Which were the inclusion criteria for healthy horses? In other words, how could the Author define their healthy status?

Lines 93-94: how could the Author define the healthy status of dogs, horses and cattle?

Statistics

I suggest to rephrase this paragraph

Results

Some results are reported in the text, in a Table (Table 2) and in a Figure (Figure 1). I suggest to delete results in the text (lines 136)

Some results are reported in the text and in a Table (Table 2). I suggest to delete results in the text (lines 139-140).

In Table 2, what does the last column (“in our study [11]) means?

Discussion

As a general comment I suggest to outline some weaknesses of the study. While some results (comparison among species) are supported by statistical analysis, others (e.g. comparison among horses) are not and sample size is limited to draw some of the conclusions.

Reviewer 2 Report

Gneral comment

The objective of the study is still unclear. Please clearly indicate what the author would like to report on.

It is still unclear why the author need to compare horse's data with dairy cattle.

One idea is to alter your title to the following

The variation of the d-ROMs and BAP concentration between horse breeds and the influence of exercise and disease.

or

Comparison of d-ROMs and BAP concentration in horse, dairy cattle and toy poodle

I also can not accept data which was used in other study (which is alreay published) to be included in the study.

Specific comments

Title: Please reconsider the title after making clear the object (hypothesis) of your study

L20-22: The author can not say that studied d-ROM level can be used to distinguish disease. As you have commented, you do not have enough data to prove that.  Also, why did the author only compared the horse data with dairy cattle and toy poodle? There should be a lot of reference (already published) related to oxidative stress in other anials including human. Please be noted that d-ROM and BAP is not the only oxidative stress marker.

L31-32: Delivery and castration are not exercise nor disease.

L33-34: If this is the object of the study, the author should not just focus on horses, dairy cow and toy poodle.

L34-35: The author do not have enough data to say that d-ROM and BAP levels can be an promising index for horse's health status.

L58: If the author want to focus on horses in this study, the author should introduce all references related to horse which are published to make the objective of the study tobecome more clear.

Materials and Methods:I also can not accept data which was used in other study (which is alreay published) to be included in the study.

Table1, 2, Figure1: The title needs to be changed (dog to poodle)since you only have data from Toy poodle

Conclusion

Is this the first study to find out that d-ROM and BAP levels differ among different animals? Although I disagree, if the author believes so, this should be the objective of your study. 

Author Response

This manuscript is a resubmission of an earlier submission. The following is a list of the peer review reports and author responses from that submission.

Round 1

Reviewer 1 Report

Major Comment

The study design is not sound. Needs clarification whether the author want to investigate possibility of using OS markers to measure health status of horses or to study the disequilibrium between oxidants and anti oxidants in horses.

Secondly, author must present details about the collected samples. Currently, it is difficult to understand when the sample was taken and stored for how long etc. These information are very important since the author may need to to consider degrading of the serum which may affect the testing result. 

Thirdly, the author can not combine all data of dogs in to one and compare with horses and cattle since the variability in dogs are too large. It may be possible to compare just Toy poodles with other animals since there are 31 Toy pooled involved.

Finally, the compared data in horses were very interesting from the clinical point. Well done.

Specific comments

Introduction

L43-50: It is unclear what author wants to say. Study on OS has been conducted for many years and there are multiple scientific papers published. Did the author wanted to state that study conducted on horses or study to compare the difference of the disequilibrium between oxidants and anti oxidants?

L51-58: These lines also needs improvement. Please clearly indicate the reasons of conducting your study. Why do you need to compare horses with other animals?

Materials and Methods

Please provide precise information of the samples used (the timing of collection, how long it has been stored? etc.)

Results

Table2:

It is not appropriate to combine all data of dogs in to one and compare with horses and cattle since the variability in dogs are too large. Unless, the author provide evidence that OS markers do not change among dog species. If there were any previous study comparing OS marks in dogs, there is no relevance I showing your data in the study.

The data is misaligned.

Discussion

L163:It is not appropriate to combine all data of dogs in to one and compare with horses and cattle since the variability in dogs are too large. 

L177-207: There is a good paper published in 2016 which reference range of blood biomarkers for OS in Thoroughbred horses. Kusano etal. J. Equine Sci-Fi. Vol.27, No.3 pp. 125-129. Strongly suggest that the author refer to that paper and improve the discussion.

Reviewer 2 Report

Introduction

The section focuses mainly on the correlation of oxidative stress with the health status/diseases, whereas  the aims of the study are:

-          comparing serum d-ROMs levels among species

-          investigating possible differences between horses (in particular in   athlete horses)

As few data concerning diseased/stressed animals are presented in the result section, I suggest to better align the introduction with the purposes of the study  

Which is/are exactely the hypothesis to be tested? Those stated in the title of the article? Effects of exercise on d-ROMs levels? To test the effects of different kind of exercise?

Materials and Methods:

As a general comment the study design is not clear.

Animals and sample preparation

Comparison among species:

Inclusion criteria for each animal species are not clear, especially considering the premises (e.g. influence of the health status/exercise on oxidative stress markers)

Comparison among horses:

Which is the study design concerning comparison among horses?

Sample origin:

If I understood well, samples were collected as follows:

Horses of the equestrian club: collected in a previous study + 8 newly enrolled horses

Horses suffering from a disease: n=3. When were they sampled?

Sport horses: data (race and endurance) were taken from previous studies and used as a model for healthy sport horses.

Data from special cases (delivery, castration): When were they sampled?

Cattle: samples collected in a previous study

Dogs: samples collected for this study

This section is quite confusing

Statistics

Please justify the use of the mentioned statistic analysis (Kruskal-Wallis test and the Dunn post test)

Results:

Some results are reported in the text, in a Table (Table 2) and in a Figure (Figure 1). The latest seems quite confusing. At the same way, Figure 2 is not clear. 

In Table 2, data related to horses are referred to horses of the equestrian club?

Discussion

The following sentence: “Clearly, because animals are not in the habit of smoking or drinking, the higher d-ROM levels in horses must be influenced by other factors. Because the amount of exercise performed by dairy cattle and dogs was thought to be obviously less than that of horses, the results suggest that the exercise amount is one reason for the higher d-ROM levels in horses” is quite speculative and not adequately supported by references

The density of oxygen-carrying erythrocytes in the blood of horses is known to be greater than in other animals such as dogs and cattle [18]”. Please check the consistency of the cited reference

Because the level of reactive oxygen species in the blood of athletes, who consume more oxygen, is invariably higher than in non-athletes, the basis of the higher d-ROM concentration in serum of horses may be for the same reason” What does the sentence refer to? Which kind of athletes?

Thus, it is possible that BAP and OSI levels decrease with age. Because the age of the horses at the  equestrian club was obviously higher than that of the dairy cattle, the higher OSI, due to lower BAP level in the serum of the horses, was thought to be age-related”. Is there the possibility, through statistical approach, to demonstrate an age-related change of this parameter?

d-ROM levels for horses having no disease but having experienced castration surgery or delivery were also distributed outside the normal range, indicating the it is not only disease that is reflected in d-ROM and BAP levels in serum”. The sentence needs to be rephrased. Moreover, BAP levels do not seem different among deseased animals (e.g. melanoma and ringbone)

Conclusions

Knowledge of d-ROMs and BAP in animals could help in studies on humans” As far as correlation between d-ROMs and disease is concerned, it seems that a lot of data in human medicine have already been obtained. Could the results be more helpful in equine sport medicine?   
